# Proteomic Characterization of Synaptosomes from Human Substantia Nigra Indicates Altered Mitochondrial Translation in Parkinson’s Disease

**DOI:** 10.3390/cells9122580

**Published:** 2020-12-02

**Authors:** Sarah Plum, Britta Eggers, Stefan Helling, Markus Stepath, Carsten Theiss, Renata E. P. Leite, Mariana Molina, Lea T. Grinberg, Peter Riederer, Manfred Gerlach, Caroline May, Katrin Marcus

**Affiliations:** 1Medizinisches Proteom-Center, Medical Faculty, Ruhr-University Bochum, 44801 Bochum, Germany; sarah.plum@rub.de (S.P.); Britta.eggers@rub.de (B.E.); Stefan.helling@rub.de (S.H.); Markus.stepath@rub.de (M.S.); caroline.May@rub.de (C.M.); 2Medical Proteome Analysis, Center for Proteindiagnostics (PRODI), Ruhr-University Bochum, 44801 Bochum, Germany; 3Department of Cytology, Institute of Anatomy, Ruhr-University Bochum, 44780 Bochum, Germany; carsten.theiss@rub.de; 4Department of Pathology, LIM22, University of Sao Paulo Medical School, Sao Paulo 01246-903, Brazil; renataparaizoleite@gmail.com (R.E.P.L.); mariana.molina01@gmail.com (M.M.); Lea.Grinberg@ucsf.edu (L.T.G.); 5Division of Geriatrics, LIM 66, University of Sao Paulo Medical School, Sao Paulo 01246-903, Brazil; 6Department of Neurology, Memory and Aging Center, University of California, San Francisco, CA 94158, USA; 7Center of Mental Health, Clinic and Policlinic for Psychiatry, Psychosomatics and Psychotherapy, University Hospital Wuerzburg, Margarete-Höppel-Platz 1, 97080 Wuerzburg, Germany; peter.riederer@uni-wuerzburg.de; 8Psychiatry Department of Clinical Research, University of Southern Denmark Odense University Hospital, Winslows Vey 18, 5000 Odense, Denmark; 9Center of Mental Health, Department of Child and Adolescent Psychiatry, Psychosomatics and Psychotherapy, University Hospital of Wuerzburg, University of Wuerzburg, 97080 Wuerzburg, Germany; manfred.gerlach@uni-wuerzburg.de

**Keywords:** synaptosomes, proteomics, Parkinson’s disease, substantia nigra pars compacta, mitochondrial pathology, mitochondrial translation

## Abstract

The pathological hallmark of Parkinson’s disease (PD) is the loss of neuromelanin-containing dopaminergic neurons within the substantia nigra pars compacta (SNpc). Additionally, numerous studies indicate an altered synaptic function during disease progression. To gain new insights into the molecular processes underlying the alteration of synaptic function in PD, a proteomic study was performed. Therefore, synaptosomes were isolated by density gradient centrifugation from SNpc tissue of individuals at advanced PD stages (N = 5) as well as control subjects free of pathology (N = 5) followed by mass spectrometry-based analysis. In total, 362 proteins were identified and assigned to the synaptosomal core proteome. This core proteome comprised all proteins expressed within the synapses without regard to data analysis software, gender, age, or disease. The differential analysis between control subjects and PD cases revealed that CD9 antigen was overrepresented and fourteen proteins, among them Thymidine kinase 2 (TK2), mitochondrial, 39S ribosomal protein L37, neurolysin, and Methionine-tRNA ligase (MARS2) were underrepresented in PD suggesting an alteration in mitochondrial translation within synaptosomes.

## 1. Introduction

Parkinson’s disease (PD) is the second most common neurodegenerative disease in the elderly [1]. Lewy body disease, the neuropathological counterpart of PD, features a preferential loss of neuromelanin-containing dopaminergic neurons in the substantia nigra pars compacta (SNpc), accompanied by the accumulation of intracellular proteinaceous inclusions named Lewy bodies and a reduction in striatal dopamine as one of its most striking findings [2]. This ongoing loss of dopaminergic neurons mainly leads to clinical diagnosis due to occurrence of motor symptoms such as rigidity, tremor and bradykinesia, which results from a reduction of about 70% of striatal dopamine [3,4].

However, neuronal loss in PD occurs in many other brain regions, including the locus coeruleus, nucleus basalis of Meynert, pedunculopontine nucleus, raphe nucleus, dorsal motor nucleus of vagus, amygdala, and hypothalamus [5]. Neurodegeneration of these neurons results in the dysfunction of other neurotransmitter systems including cholinergic, GABAergic, glutamatergic and noradrenergic neurotransmission [6]. That leads to a wide range of non-motor symptoms that sometimes even precede the typical movement disorder, such as olfactory dysfunction, sleep disturbances (i.e., rapid eye movement sleep behavior disorder), psychiatric symptoms such as depression, autonomic dysfunction, pain, and fatigue [7]. The causes of these neuronal losses have hardly been investigated so far. Many of these neurons project into the SN and are thus probably involved in the neurodegenerative processes there.

Synaptosomes represent synapses detached during tissue lysis from neuronal cell bodies [8,9] and still contain the machinery required for signal transduction [10]. The study of synaptosomes provides insights into the physiological and pathophysiological processes in neurotransmission and synaptic protein–protein interaction networks [11]. Differences in the protein composition of synaptosomes in the context of a disease might indicate a disturbed signal transduction in the synapses. Pathological deficits in the synaptic processes in neurological disorders have been studied in the past especially for Alzheimer′s [12,13,14,15], Huntington′s disease [16], Lewy body disease [17] and PD [18] as well as schizophrenia [19,20,21]. Omics approaches including proteomics are promising new avenues to study synaptosomes [22,23]. Indeed, human synaptosomes have so far only been investigated in a handful of proteomics studies [24,25] mainly in Alzheimer’s disease [10,26,27], schizophrenia [28,29] and only one investigates pathophysiological molecular processes in Lewy body disease (LBD) [30].

To elucidate a possible pathophysiological effect leading to PD caused by a loss of neurons that project into the SN, we analyzed the proteome of synaptosomes from human postmortem SNpc tissue. In the past, different protocols for the enrichment of synaptosomes by, e.g., density gradient centrifugation, have been established [31] (for review see [32]). Here, a combined approach of a protocol designed by Dunkley et al. [33] with a strategy to enrich neuromelanin granules allowing for a case-specific enrichment [34] was used. Differences in the synaptosomal proteome of PD subjects and controls were identified by bottom-up liquid chromatography tandem mass spectrometry (LC-MS/MS). From these results, a synaptosomal core proteome of the SNpc could be derived for the first time. Disease-related differences were detected, indicating an altered mitochondrial translation in synaptosomes of individuals suffering from PD.

## 2. Materials and Methods

### 2.1. Ethical Statement

Human postmortem SNpc tissues were provided by the German Brain Bank in Wuerzburg, Germany, and the Brazilian Brain Bank in São Paulo, Brazil. The use of human brain tissue was approved by the ethics committee of the University Clinics of Wuerzburg, Germany (file number 78/99), the ethics committee of the Ruhr-University Bochum, Germany (file number 4760-13), the ethics committee of the University of São Paulo (file number 361/10), and the Brazilian national health ministry (file number 16380).

### 2.2. Study Cohort for Subsequent Proteomic Analyses

Details about the analyzed study groups are shown in Table 1. For all PD cases, the neuropathological severity of the disease was assessed according to measures defined by Braak et al. and ranged between stages 3 to 6 [35].

### 2.3. Tissue and Sample Preparation for LC-MS/MS Experiments

All subjects died of natural causes, and their brains were dissected and stored at −80 °C, as previously described [36]. A detailed description of the respective procedures can be found elsewhere [37]. Brain tissue was obtained within 24 h of death. One hemisphere was fixed in 4% buffered paraformaldehyde, and selected brain areas from the other hemisphere were frozen at −80 °C. Internationally accepted neuropathological criteria were used to stage and diagnose the brain pathologies [35,38,39,40,41]. The neuropathological term Lewy body disease (LBD) was used for all diseases associated with Lewy bodies, thereby eliminating the distinction between PD, PDD (Parkinson’s disease dementia) and DLB (dementia with lewy bodies). We staged LBD using the criteria of Braak and colleagues. For sample characteristics, see Table 1.

Synaptosomes were enriched from PD and control SNpc tissues (each N = 5) with centrifugation as described [34]. Pelleted synaptosomes were dissolved and lysed by addition of 20 µL deionized water, resuspension with a pipette five times, and then vortexed. Protein concentration was determined by amino acid analysis [34]. Then, 5 µg of each sample was diluted with deionized water to an end volume of 14 µL. Then, 3.5 µL lithium dodecyl sulfate (LDS) buffer (4x-buffer: 106 mM Tris HCl (Sigma Aldrich GmbH, Hofheim, Germany), 141 mM Tris base, Sigma Aldrich GmbH; 2% (*w/v*) LDSAppliChem GmbH, Darmstadt, Germany; 10% (*w/v*) glycerol, Sigma Aldrich GmbH; 0.5 mM EDTA, Merck KGaA, Darmstadt, Germany) and 1.75 µL 2 M dithiothreitol were added to each sample and samples were heated for 10 min at 95 °C. Samples were run in short 12% acrylamide BisTris gels and tryptic digestion occurred over-night. Extracted peptides were taken up in 30 µL 0.1% trifluoroacetic acid (TFA) and peptide concentration was determined by amino acid analysis [34].

### 2.4. Electron Microscopy

Electron microscopic analysis was performed as previously described [34]. Briefly, sediments were incubated overnight in 2.5% glutaraldehyde (Fluka, Sigma-Aldrich Chemie GmbH, Hofheim, Germany), washed with phosphate buffered saline, fixed with Dalton fixing solution, embedded in Epon 812, dehydrated and counterstained. Subsequently, samples were dehydrated with an ascending ethanol series and transferred to Epoxypropane. Samples were transferred into Epon 812 by incubation with a series of increasing amounts of Epon 812 in Epoxypropane. Then, samples with hardener were cast in stencils and after two days of polymerization at 60 °C, blocks were sliced in 70 nm sections using an ultramicrotome. Sections were mounted on formvar coated 75 mesh copper grids and analyzed using a transmission electron microscope (Philips-420, Philips, Hamburg, Germany) with a digital camera (Gatan, Inc., Pleasanton, CA, USA).

### 2.5. Mass Spectrometric Analysis

Prior to mass spectrometric analysis, 200 ng peptides per case were loaded on a capillary pre-column (Dionex, 100 µm × 2 cm, particle size 5 µm, pore size 100 Å), that was washed for 7 min with 0.1% TFA. The peptides were eluted from the precolumn to an analytical C18 column (Dionex, 75 µm × 50 cm, particle size 2 µm, pore size 100 Å). Peptide separation was performed with a gradient that began with 95% A (0.1% formic acid, Fluka, Sigma-Aldrich Chemie GmbH) and 5% B (84% acetonitrile (J.T. Baker^®^, Avantor Performance Materials, Inc., Center Valley, PA, USA oder “LC/MS grade“, Biosolve b.v., Valkenswaard, Niederlande), 0.1% formic acid (Fluka, Sigma-Aldrich Chemie GmbH) with a 400 nL/min flow rate. The concentration of B was increased up to 40% within 98 min, then increased to 95% within 2 min and maintained for 3 min. Afterwards, the column was again adjusted to 5% B. The nano HPLC system (Thermo Fisher Scientific Inc., Waltham, MA, USA) was directly coupled with a nano-electrospray ionization source (Thermo Fisher Scientific) to the Orbitrap Elite mass spectrometer (Thermo Fisher Scientific Inc.). The system operated with a scan range from 300 to 2000 *m/z* with a resolution of 30,000 and 500 ms maximum acquisition time. From each full scan, the 20 most intensive ions were selected for low-energy collision-induced dissociation (CID) with 35% collision energy and 50 ms maximal acquisition time. After fragment ion (MS/MS) scans, the mass to charge (*m/z*) values of the precursor masses were maintained for 30 sec on a dynamic exclusion list.

For quality control during the analysis, standard samples of tryptic digested A549 cells were measured in between the samples. Master mixes including same the amounts of each analyzed sample were measured at different time points at the beginning, during, and at the end of the analysis to be able to estimate technical variances of LC-MS/MS analyses.

The mass spectrometry raw data were submitted to the ProteomeXchange Consortium via the PRIDE partner repository [42] with the dataset identifier PXD022092. 

### 2.6. Data Analysis

Data resulting from LC-MS/MS measurements were subsequently analyzed with different software aided strategies. First, spectra were matched to peptides, using the Mascot algorithm. Protein inference and quantification were carried out using Protein Inference Algorithm (PIA) [43,44] in an automized spectral counting workflow implemented in KNIME as previously described [45,46]. Second, identification was performed using the Andromeda algorithm integrated into MaxQuant (MQ) and quantified using the label free quantification (LFQ) function as well as the intensity based absolute quantification (iBAQ) option, estimating absolute protein abundances by using the raw intensities divided by the number of theoretical peptides, resulting in an estimation of their absolute concentration [47,48]. In both approaches, spectra were matched to in silico generated spectra resulting from the protein information deposited in a UniProt/SwissProt decoy database (release 2020 from 07.10.2020 20,370 protein entries without decoys). Fragment mass tolerance was set to 0.4 Da and the enzyme was set to “trypsin”, with a maximum of two missed cleavages accepted. Oxidation at methionine and deamidation at asparagine and glutamine were set as a dynamic modification. All further settings (like FDR <0.01) were used as according to the MQ default settings. LFQ values represent normalized summed up spectra intensities for peptides connected to a protein identified in a sample. Further data processing was carried out using Perseus. Prior to statistical evaluation with a *t*-test, the LFQ values were transformed by calculating their logarithms to the base of two. As values equal to 0 could not be transformed, these were set to 0 manually. Student’s *t*-test was calculated and proteins with a *p*-value < 0.05 and present in all samples were selected as being differentially expressed between controls (CTRL) and PD cases. Fold changes were estimated by calculating the quotients of the mean LFQ values for neuropathologically healthy and PD cases. If these values were lower than 1, negative inverse values were calculated.

### 2.7. Analysis of Protein Annotations

Resulting protein lists were subjected to different annotation tools for further in-depth analysis. For characterization of the synaptosome core proteome, protein lists were subjected to annotation clustering using Bionic Visualization′s Proteomaps (https://www.proteomaps.net). This enabled a visual supported data evaluation [49,50]. Additionally, data were enriched with information deposited in the Gene Ontology database (http://GeneOntology.org/page/go-database). For annotation enrichment of differential protein lists, the web version of DAVID Bioinformatics Resources 6.8 was used [51,52]. Proteins were assigned to Gene Ontology terms as before with regard to biological processes.

### 2.8. Western Blotting

After amino acid analysis, 10 µg of sample was diluted with deionized water to have equal amounts of sample volumes. Then, LDS buffer and 2 M dithiothreitol (1:10 *v*/*v*) were added to the samples, and samples were heated for 10 min at 95 °C. Samples were loaded together with a marker (PageRuler™ Prestained Protein Ladder or Novex^®^ Sharp Pre-Stained Protein Standard, Thermo Fisher Scientific) to a 4–12% acrylamide BisTris gel. Gels were run with MES buffer for 15 min at 50 V and 40 min at 160 V. Proteins were then transferred to nitrocellulose membranes using the iBlot^®^ system (Thermo Fisher Scientific) and membranes were blocked for 30 min with StartingBlock (TBS) Blocking Buffer (Thermo Fisher Scientific). After washing with TBS (25 mM Tris-Base, 73 mM NaCl, 2.7 mM KCl, pH 8.7) for 10 min to remove excess blocking buffer, membranes were incubated over-night with different antibodies diluted in TBS (Rabbit anti-Synaptophysin, SAB4502906, 1:500, Sigma-Aldrich Chemie GmbH, Taufkirchen, Germany; Rabbit anti-GFAP, PAB12119, 1:50,000, Abnova Corporation, Taipei, Taiwan; Mouse anti-AIF-1, HM2184, 1:500, Hycult Biotech, Uden, The Netherlands). Then, membranes were washed three times, for 10 min each, with TBS-T (0.1% Tween 20 in TBS) and incubated for one hour with a fluorophore coupled secondary antibody suitable to detect the first antibody, diluted 1: 15,000 in TBS IRDye^®^ (800CW or 680RD Goat anti-Rabbit or anti-Mouse IgG (H + L), LI-COR Biosciences GmbH, Bad Homburg, Germany). Membranes were again washed three times with TBS-T, followed by another TBS wash (three times) for 10 min each. Subsequently, signals were detected using the Odyssey classic Imaging system and visualized with the Image Studio ™ Light software provided by the manufacturer (LI-COR Biosciences GmbH, Bad Homburg, Germany).

## 3. Results

### 3.1. Western Blot Analysis Identifies a Sub-Fraction Suitable for In-Depth Proteomic Analysis

The workflow of the represented study is schematically summarized in Figure 1. SN was dissected from fresh frozen postmortem human midbrain. Then the midbrain of one hemisphere of each case was lysed using a Dounce homogenizer and insoluble fragments were removed by a short centrifugation step. The as-generated supernatant was further fractionated by density gradient centrifugation to enrich synaptosomes. Using electron microscopy and Western blot analysis, interphase 15 to 23% Percoll was identified as the fraction with the highest purity of synaptosomes and therefore taken for in depth proteomic analysis by nanoLC-MS/MS. Two different algorithms—Mascot and Andromeda, were used to assess the resulting spectra. The results were compared and proteins that were identified with both algorithms in all samples were declared as the core proteome of SN synaptosomes. Furthermore, a differential analysis was performed between controls and PD cases using spectral counting and MQ label free quantification (LFQ). Again, resulting candidates were compared leading to the identification of differential candidates. As shown in Figure 2, no organelle contaminations like nuclei or Golgi stacks could be observed after the purification based on density gradient centrifugation. At 10,500-fold magnification (Figure 2B) synaptosomes (examples indicated by arrows) are clearly detected. With higher resolution, mitochondria can be identified in synaptosomes (indicated by asterisk in Figure 2C) resembling morphologically functional synapses.

As it was not clear if protuberances of astrocytes or microglia were co-enriched in synaptosomes samples, all fractions of a SNpc tissue extracted from one control case were subjected to Western blot analysis. As demonstrated in Figure 2D, signals for GFAP, a marker for astrocytes, were strong in lysate, supernatant, or pellet fractions, but signals decreased in fractions of the differential Percoll gradient. No GFAP-signal was detected in interphase 15/23% Percoll. A marker for activated microglia, AIF-1, was also tested. Results are shown in Figure 2D. While a strong signal occurs in interphase 0/3% Percoll, weaker signals were detected in lysate supernatant, pellet, and interphases 3/10% as well as 10/15% Percoll. No signal could be observed in interphase 15/23% Percoll. Additionally, anti-synaptophysin antibodies were applied to identify synaptosomes containing fractions as shown by DiGiovanni et al. [53]. As can be seen in Figure 2D, synaptophysin signals were clearly detectable in all Percoll gradient fractions, with the strongest signals in interphase 10/15% and 15/23% Percoll. Data showed that synaptosomes are strongly enriched in the interphase 15/23% Percoll, while no signals of the markers for astrocytes (GFAP) and microglia (AIF-1) were identifiable in this fraction. Thus, this fraction contains synaptosomes with the highest purity and was used for further experiments.

To ensure a similar abundance of mitochondria within each sample, we additionally included intensity based absolute quantification (iBAQ) [54] values as an option in our MQ analysis. As iBAQ values are calculated using the raw intensities divided by the number of theoretical peptides, they are proportional to the molar quantities of each individual protein, resulting in an estimation of their absolute concentration. To validate similar amounts of mitochondria ATP-Synthase and the mitochondrial TIM/TOM complexes abundances were assessed. For that, iBAQ values of proteins forming the ATP-Synthase (17 proteins) complex and the TIM/TOM complex (18 proteins) were calculated into percentage values and summed up for each sample separately resulting in a percentage for both mitochondrial marker protein complexes (Table 2 & Appendix A).

### 3.2. Core Proteome of Synapses Isolated from the Substantia Nigra

Synaptosomes of all ten individual human SNpc samples without regard to age, gender, or disease, (Table 1) were analyzed by LC-MS/MS, and proteins were quantified using two different search algorithms (Mascot and Andromeda) (for an overview see Figure 1). This strategy allows cross-validation of mass spectrometric results. In total, 3125 proteins were quantified with Mascot (Appendix A) and 2764 with Andromeda (Appendix A). In combination, 2544 protein groups were quantified with both algorithms. According to our quality requirements, 362 of 2544 proteins could be quantified in every sample. These 362 proteins represent the proteins expressed in human synaptosomes isolated from the SNpc without regard to age, gender, disease, or algorithm and are declared as the core proteome (see Appendix A). The top 50 proteins were ranked regarding their intensity (LFQ intensity MQ) for CTRL and PD cases separately and were plotted to verify abundance differences. Most proteins (n = 38) showed an identical or similar ranking in both CTRL and PD cases. Proteins being in the top 50 group only were marked in red (Figure 3). As our rank–intensity plots showed comparable patterns for both cases, one could summarize that these proteins are expressed in comparable amounts and seem to be characteristic for synaptosomes of human subjects of the SNpc.

### 3.3. Functional Annotation of the Core Proteome

The core proteome of synaptosomes isolated from the SNpc (362 proteins) was subjected to an annotation clustering using Bionic Visualizations Proteomaps [49,50]. Here, functional information was drawn from terms describing the proteins.

The terms were clustered in six basic categories identifiable by color-coding (Figure 4A): human diseases (black), environmental information processing (cyan), genetic information processing (blue), metabolism (yellow), cellular processes (red), and organismal systems (pink). Most proteins of the core proteome list are assigned to a certain category, the larger area of this category is shown in the figure. As seen in Figure 4, many proteins are associated with metabolism (yellow/brown, Figure 4A), especially biosynthesis, central carbon metabolism, as well as energy metabolism (yellow/brown, Figure 4B). A more in-depth analysis reveals that proteins associated with metabolism represent the amino acid metabolisms, glycolysis, oxidative phosphorylation, lipid/steroid metabolism and other enzymes (Figure 4C). Cytoskeleton protein, endocytosis, lysosome, tight junction and cell cycle (Figure 4C) represent cellular processes (red, Figure 4A) especially vesicular transport (Figure 4B). Main components of the environmental information processing (turquoise area, Figure 4A)/ signal transduction (Figure 4B) are part of the MAP (mitogen-activated protein)-kinase, calcium and Rap1 signaling pathways and cell adhesion (Figure 4C). Regarding organismal systems (pink), the immune system, the circulatory system and nervous system were represented by most of the proteins in the list (Figure 4B). Most of the proteins here were assigned to cardiac muscle contraction and Fc gamma R-mediated phagocytosis (Figure 4C). Indeed, the grouping does not seem to be very expressive here, as there are many small areas in this pink part. Finally, genetic information processing (blue area, Figure 4A) especially particularly folding, sorting, degradation, and translation (Figure 4B) is represented by chaperons/folding catalysts and proteins involved in processing in the ER (endoplasmic reticulum), tRNA loading and mitochondrial biogenesis (Figure 4C). Further, in the area of genetic information processing (blue), terms like SNARE interaction, in vesicular transport, and protein processing can be seen (Figure 4C, black stars). Detailed information regarding which proteins were annotated in each cluster can be found in Appendix A. In summary, many terms suggest typical synaptic processes emphasizing the eligibility of the defined synaptosome core proteome.

### 3.4. Proteins Altered between Parkinson’s Disease Patients and Control Subjects

To find differences in the expressed proteins, synaptosomes from controls and PD patients were analyzed and relatively quantified using two different analysis strategies in parallel: spectral counting based on peptide spectrum matches (PSMs) of unique peptides and intensity based analysis with the label-free quantification (LFQ) function of MQ. Proteins were accepted as being differentially expressed with a *p*-value ≤ 5%. Using spectral counting, 103 proteins fulfilled these criteria; using MQ, 73 proteins were accepted (Appendix A). Comparison of the two data analysis strategies revealed 15 proteins that were regulated in both data analysis strategies (see Table 3). These proteins represent differential proteins with strong evidence.

One protein was significantly overrepresented (italic) and fourteen proteins were underrepresented. Eleven underrepresented proteins were found to be located in the mitochondrion, either associated with energetic processes, such as the fatty acid beta oxidation, the citric acid cycle or mitochondrial translation pointing to an alteration in mitochondrial function in PD synaptosomes. Mass spectrometric data are publicly available for further data analysis in PRIDE with the dataset identifier PXD022092.

Again, we additionally investigated protein abundances using the MQ intern iBAQ approach. Proteins being significantly overrepresented in the CRL group showed enhanced iBAQ values and when compared to higher PD ratios (Appendix A), support our relative quantification approach. 

### 3.5. Confirmation of Altered Mitochondrial Translation in PD Synaptosomes

In order to support and consolidate our findings of an altered mitochondrial translation in PD-affected synaptosomes, we extended our investigations by including the individual lists of differential proteins from spectral counting and MQ for further GO term enrichment analysis using DAVID Bioinformatics Resources 6.8 (Appendix A). Resulting terms were grouped according to biological process, as described in Material and Methods, resulting in graphs as shown in Figure 5. Our GO term enrichment analyses additionally underline that proteins being significantly enriched in CTRLs are connected to mitochondria ribosomes, or translational processes. In particular the terms “mitochondrial translation elongation” as well as “mitochondrial translation termination” reached the highest significance with *p*-values < 1 × 10^−11^ and fold enrichment scores > 32, supporting our original hypotheses.

## 4. Discussion

Using density gradient centrifugation, we were able to isolate pure synaptosomes from SNpc tissue of controls and PD patients. A qualitative proteome analysis revealed a total number of 2873 proteins in both study groups. A subsequent differential quantitative analysis between controls and PD patients yielded eight differential proteins. Detailed analysis of Gene Ontology terms and publications describing these proteins demonstrated a strong connection to an altered mitochondrial translation in PD.

### 4.1. Isolation of Synaptosomes from and Definition of Synaptosome Core Proteome in the Substantia Nigra

Synaptosomes from SNpc tissue of controls as well as PD patients were purified using a Percoll density gradient. Using transmission electron microscopy and Western blot analysis, we could confirm that the gradient fraction 15/23% Percoll contained synaptosomes with the highest purity. No organelles, astrocytes and microglia were found in this fraction and synaptosomes resembled morphologically functional synapses. Hence, this fraction was further used for in-depth proteome analysis of PD as well as control case synaptosomes by mass spectrometry. In the first instance, a so-called core proteome was defined. This core proteome includes all proteins expressed within the synapses without regard to data analysis software, gender, age or disease (see Figure 1).

In total, 363 proteins were identified and defined with strong evidence using two different algorithms as synaptosome core proteome. A table of this core proteome can be found in the Appendix A and additionally the raw data were uploaded to PRIDE with the dataset identifier PXD022092. Using a different annotation strategy, we found that these proteins are associated with synaptic processes like, e.g., signal transmission and vesicle cycle, demonstrating a high quality of our fractionation protocol and analysis strategy.

### 4.2. Identification of PD-Related Changes

We compared the protein profiles of PD and control synaptosomes to detect potentially pathological molecular mechanisms underlying neurodegeneration in PD. Fifteen proteins were identified; one of them was more highly expressed in PD synaptosomes compared to controls. In total, 14 were expressed at low levels in PD. Among them: Thymidine kinase 2 (TK2), 39S ribosomal protein L37, neurolysin, and Methionine-tRNA ligase (MARS2). Interestingly, all these proteins are integral parts of biosynthetic pathways and most of these proteins were not mentioned before in the context of PD.

CD9 antigen was the only protein that was more highly expressed in PD synaptosomes compared to controls. In the Gene Ontology database, CD9 antigen is inter alia connected to brain development, signaling, and endocytic vesicles. CD9 belongs to the tetra-membrane-spanning protein family together with TAPA-1, CD37, CD53, and CD63. The function of these proteins is still not well understood. Originally, CD9 was identified as surface antigen of lymph hemopoietic cells [55], later it was found to be component of the myelin in the central nervous system as well as the peripheral nervous system [56]. Schenk et al. showed in 2013, that blocking of CD9 significantly reduced the migration of monocytes across brain endothelial cell monolayers and enhanced the barrier function of the blood brain barrier in vitro [57]. Neuroimaging studies revealed an early blood brain barrier dysfunction in PD and hypothesized that it contributes to PD pathology [58]. Therefore, it can be assumed that blocking of CD9 might also enhance blood brain barrier function in the context of PD and might positively influence PD pathology.

### 4.3. Changed Mitochondrial Translation and mtDNA Synthesis in PD

Mitochondrial dysfunction is a central aspect of aging and neurodegenerative diseases, including Alzheimer’s disease and PD. Already in the 1980s, evidence for linkage of mitochondrial dysfunction and PD was found when accidental exposure to 1-methyl-4-phenyl-1,2,3,6-tetrahydropyridine (MPTP), a contaminant from the synthesis of 1-methyl-4-phenyl-4-propionoxy-piperidine (MPP+) (a drug used for illicit purposes and mitochondrial complex I inhibitor), was found to cause Parkinsonism and DA neurodegeneration [59]. At the end of the 1990s a genetic linkage was also found when a mutated gene-Parkin2-coding for a mitochondrial E3 ubiquitin ligase was found to cause juvenile Parkinsonism [60,61,62]. A second gene to be identified in early-onset recessive PD was found at the PARK6 locus encoding PINK1 a mitochondrial protein phosphatase and tensin homolog (PTEN)-induced kinase 1 [63,64]. Both are mediators of mitophagy and a loss of function of Parkin and/or PINK1 may cause an accumulation of dysfunctional mitochondria leading to early onset-PD [65]. In general, three main abnormal mitochondrial processes involved in the development of neurodegeneration have been supposed, namely, impaired mitochondrial (1) dynamics (i.e., fusion and fission), (2) kinetics (i.e., transport and distribution) and (3) bioenergetics of the electron transport chain and TCA (tricarboxylic acid) cycle [66]. Since 1980, many different research groups evaluated alterations in mitochondria in the context of PD on DNA as well as protein levels in humans, cell culture as well as animal models using different approaches and techniques, including proteomics [67,68,69,70,71,72,73,74,75,76]. Our work confirms these findings and further expands the knowledge of the essential role of mitochondria in PD. Here for the first time we could clearly show that alterations in mitochondrial function in synaptosomes of the substantia nigra may contribute to the disease:

Mitochondrial 39S ribosomal protein L37 and MARS2 are proteins, which are linked to mitochondrial translation. TK2 plays a role in DNA synthesis. Since 1989, different observations support the thesis that impaired mitochondrial function is a core event underlying the pathogenesis in PD. For example, respiratory chain complex I activity and aldehyde dehydrogenase 1 activity as well as expression is reduced [60,61,62,77,78,79,80], whereas mtDNA deletions accumulate in the SN of PD subjects [81]. Moreover, several of the genes associated with familial forms of PD control mitochondrial functions by the regulation of oxidant defenses, mitophagy or biogenesis [82]. An important role of mitochondria in progressive neurodegenerative disorders such as PD was summarized by Schapira years ago [83] as well as by Obeso et al. more recently [84]. Our results support the thesis of an impaired mitochondrial function in the pathogenesis of PD.

*Thymidine kinase 2 (TK2)*—TK2 is a mitochondrial protein essential for building up mtDNA. This occurs within a salvage pathway meaning that the mtDNA building underlies a recycling process. Within this pathway, TK2 phosphorylates the nucleosides deoxythymidine, deoxycytidine as well as deoxyuridine so that they can again be incorporated into the mtDNA. Mitochondria normally contain between 10 and 15 DNA molecules, each cell up to 10,000 mtDNA copies. The mtDNA code includes 37 genes of which 13 encode for respiratory chain proteins, the remaining are two rRNA and 22 tRNA genes. A deficiency in TK2 leads to a progressive depletion of mtDNA resulting in a so-called mitochondrial DNA depletion syndrome (MDDS) when the mtDNA drops below a significant level [85,86]. The affected individuals die normally during early childhood. Most affected are tissues with a high-energy demand, in TK2 based MDDS muscles and to a less extend the brain as well as the liver tissue. Nicotera et al. evaluated the role of TK2 on neuronal homeostasis using a knockout mouse model [87]. The authors could clearly show a reduced mtDNA copy number in the whole brain, decreased levels of electron transport chain proteins in specific brain regions, and moreover within individual neurons. Additionally, a reduction in the number of dendrites and a decreased dendritic arborization was demonstrated. A reduced TK2 level in mitochondrial synapses within the SN of PD patients might have similar effects on mtDNA. It can be assumed that the respiratory chain and therewith synaptic as well as neuronal viability might be negatively affected by TK2 deficiency.

*39S ribosomal protein L37*—39S ribosomal protein L37 is a component of the large subunit (39S) of the mitochondrial ribosome (55S) [88], builds a ~100-kD heterodimer with mS30 protein and seems to be involved in the forming the mRNA channel in the mito ribosome [89]. The exact function of L37 has not yet been clarified. Genetic variants of L37 and PARK2 genes have been shown to be associated with altered levels of mtDNA in a sex-specific manner in recurrent venous thromboembolism [90]. A reduced abundance of 39S ribosomal protein L37 in PD may directly affect mitochondrial protein synthesis, which may lead to a decreased activity of diverse respiratory chain enzymes, because they are synthesized at the mitochondrial ribosomes.

*Methionine-tRNA ligase (MARS2)*—MARS2 belongs, like DARS2, to the group of mitochondrial aminoacyl-tRNA synthetases. Like DARS2 also MARS2 is encoded by nuclear genes. Rearrangements in MARS2 gene were shown to cause neurodegeneration in autosomal recessive Spastic Ataxia with leukoencephalopathy (ARSAL) [91]. In ARSAL patient cells, reduced levels of MARS2 protein, a reduction in mitochondrial translated proteins, a reduced complex I activity, increased reactive oxygen species, and a slower cell proliferation rate within patient cells were detected. Our hypothesis is that like ARSAL, a reduced MARS2 level in PD leads to neurodegeneration due to the above described processes.

*Neurolysin, mitochondrial*—Neurolysin is an endooligopeptidase localized in the cytoplasm, in the mitochondrial matrix, but was also found to be membrane associated [92,93]. It was first described by Checler et al. in 1986, who purified neurolysin out of synaptic membranes in rat brain and demonstrated that it can cleave neurotensin in two distinct inactive peptides [94]. Nowadays it is known, that neurolysin is highly abundant in the brain, including the ventral midbrain, olfactory bulb and tubercle, cingulate cortex, neostriatum, and globus pallidus, but is ubiquitously expressed in mammalian tissues [93]. Fine structural distribution analysis revealed that neurolysin can be detected throughout the perikarya, dendrites, within axons as well as axon terminals [95]. In all neuronal compartments, neurolysin additionally showed a major association with membranes of neurosecretory elements, e.g., synaptic vesicles [83]. Based on this knowledge, it can be hypothesized, that a reduced abundance of neurolysin in synaptosomes of PD patients may somehow influence neurosecretory pathways, which could negatively influence neuronal functionality in PD. Moreover, neurolysin was shown to degrade mitochondrial presequence peptides and other fragments up to 19 amino acids [96]. Teixeira et al. could demonstrate that neurolysin cooperates in vitro with mitochondrial presequence proteases in the degradation of long targeting peptides as well as amyloid peptides, also cleaving the hydrophobic fragment Aß35-40 [92]. From these results, it can be concluded that a deficient clearance of peptides by a reduced level of neurolysin, as found in our study, led to protein aggregation in the mitochondria as well as in the whole neuron, which could perhaps negatively influence mitochondrial and, furthermore, neuronal viability.

## 5. Conclusions

In summary, we isolated, for the first time, synaptosomes from the human SNpc and characterized their protein levels by mass spectrometry in controls and PD patients. By comparing the protein profiles of PD and control synaptosomes we identified 14 mitochondrial proteins that all are expressed at lower levels in PD. Our results support former findings on impaired mitochondrial function in PD [77,78,81,82,83,84] extending them by the hypothesis of altered mitochondrial translation. Concretely, the identified protein Thymidine kinase 2 (TK2), 39S ribosomal protein L37, neurolysin, and Methionine-tRNA ligase (MARS2) could be additional key players in those pathogenic processes contributing to mitochondrial dysfunction in PD. Further studies in larger and independent cohorts are necessary to confirm our findings. Furthermore, our results provide an important basis for further studies in suitable cell or animal models to better understand the functional importance of these proteins for disease pathogenesis.

## Figures and Tables

**Figure 1 cells-09-02580-f001:**
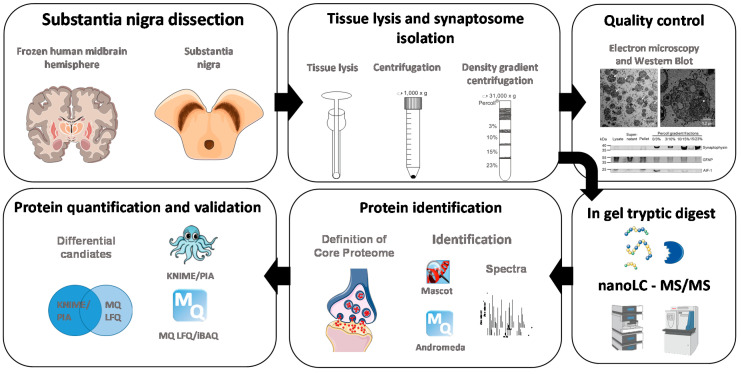
Strategy for the characterization of synaptosomes. Substantia nigra pars compacta (SNpc) was dissected from fresh frozen postmortem human midbrain tissue. The SNpc of one midbrain hemisphere of each case was lysed separately and the Dounce homogenizer was cleaned before each tissue lysis. Lysates were centrifuged separately, and supernatants were fractionated by separate density gradients. With electron microscopy and Western blot, interphase 15 to 23% Percoll was identified as the fraction with the highest purity regarding synaptosomes. Hence, synaptosomes enriched in this fraction were digested in gel by trypsin and resulting peptides were fractionated and analyzed by nanoLC-MS/MS. Resulting spectra were assessed by two different algorithms—Mascot and Andromeda. Resulting identifications were compared and only proteins that were identified with both algorithms in all analyzed samples were accepted as part of the core proteome in synaptosomes isolated from the SNpc. Further, identified proteins were quantified using spectral counting and MaxQuant (MQ) label free quantification (LFQ). Resulting candidates were compared leading to the identification of unique candidates.

**Figure 2 cells-09-02580-f002:**
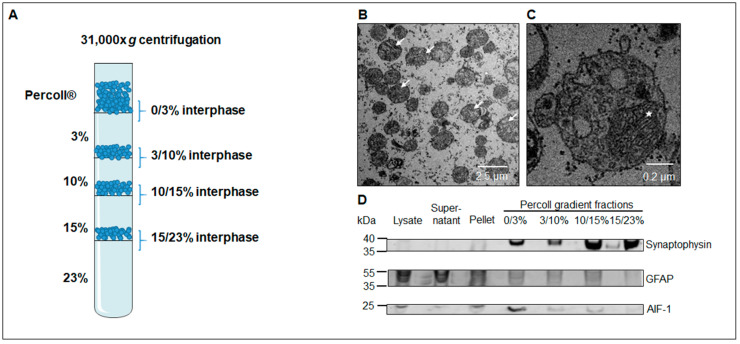
Quality control of synaptosomes enrichment. (**A**) Graphical image of the Percoll gradient for synaptosome enrichment. (**B**,**C**): Transmission electron microscopic analysis of density gradient fraction 15/23% Percoll at 10,500× (**B**) or 60,000× (**C**) magnification. At 10,500× magnification, no organelle contaminations could be observed but enrichment of synaptosomes (arrows) was obvious (**B**). Analysis at higher magnification showed synaptosomes contained mitochondria (asterisk) besides vesicles, underlining that synaptosomes resemble synapses. (**D**) Western blot analysis of different Percoll gradient fractions. Percoll fraction 15/23% did not contain astrocytes (indicated by GFAP) or microglia (indicated by AIF-1) and was identified as the purest synaptosomal fraction for further analysis.

**Figure 3 cells-09-02580-f003:**
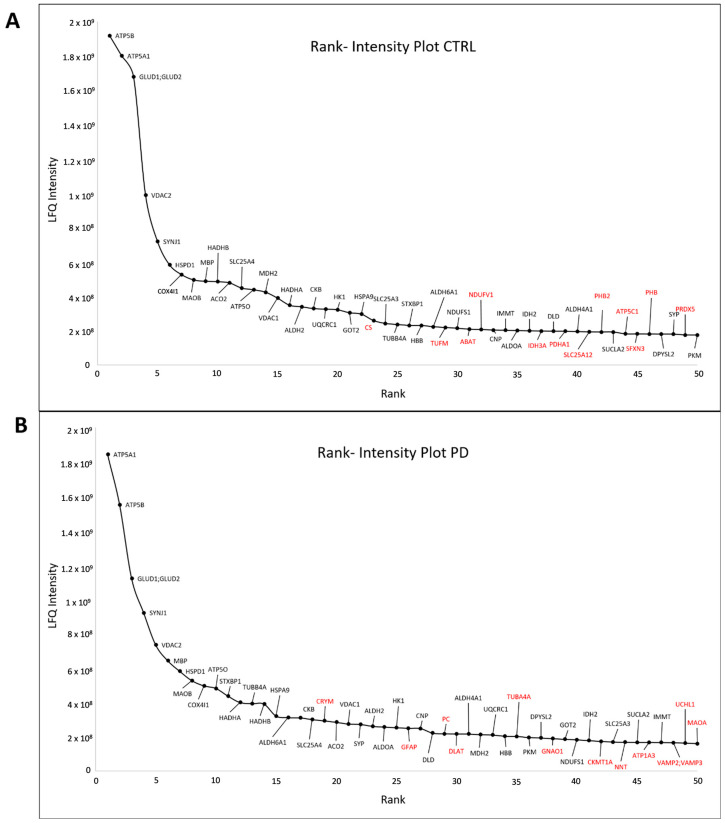
Rank–intensity plot for (**A**) CTRL and (**B**) PD. The mean of LFQ intensities for each protein in each group was calculated. Proteins were subsequently ranked from highest to lowest LFQ intensity. The top 50 proteins were plotted against their intensities. Each protein is marked using the gene name. Proteins being in the top 50 in one group only (either CTRL or PD cases) are marked in red.

**Figure 4 cells-09-02580-f004:**
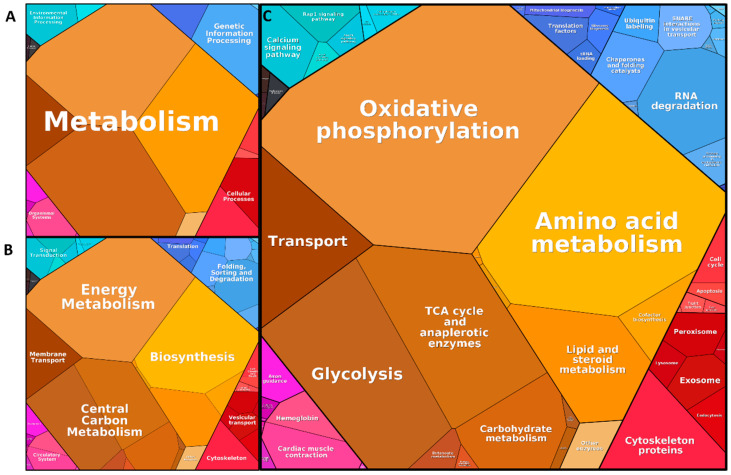
Results of the annotation clustering by Bionic Visualizations Proteomaps. Different levels of information are shown, with a higher level meaning a more in-depth analysis: in (**A**), level 1 clustering; in (**B**), level 2 clustering, and in (**C**), level 3 clustering is shown. The larger the area the more proteins are assigned to the respective category. Terms are clustered in six main categories, as can be seen in (**A**), indicated by color: human diseases (black), environmental information processing (cyan), genetic information processing (blue), metabolism (yellow), cellular processes (red), and organismal systems (pink). Many proteins of the core proteome are connected in the yellow coded area metabolism (**A**). In level 2 (**B**), biosynthesis is prominent, while more in-depth analysis at level 3 (**C**) shows strong connection to terms like amino acid metabolism, lipid and steroid metabolism, and cofactor biosynthesis (for further details see Appendix A).

**Figure 5 cells-09-02580-f005:**
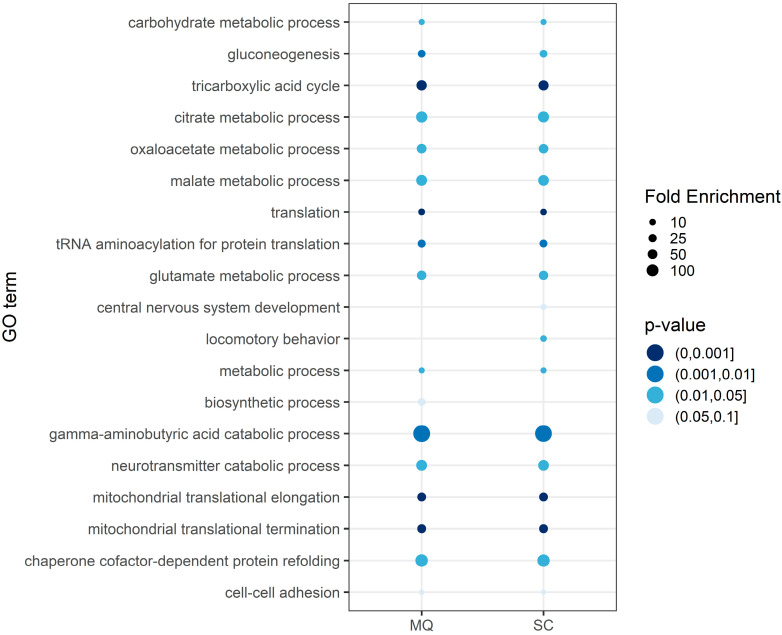
Results of the Gene Ontology term analysis (Biological Process) using DAVID Bioinformatics Resources 6.8 based on significantly differential expressed proteins between CTRL and PD in both data analysis strategies (MQ and spectral counting). Significantly enriched proteins in both strategies were verified as being part of essential mitochondrial processes, such as mitochondrial translation, the tricarboxylic acid cycle and gluconeogenesis.

**Table 1 cells-09-02580-t001:** Characteristics of the study groups.

Group	Gender	Age (in years)	Ø Age ± SD	DS	PMI (h)	Ø PMI (h) ± SD	Tissue Weight (g)	PD Braak/ AD Braak	CDR	CERAD
Control	Female	88	78.8 ± 6.9	NA	~13:00	14:04 ± 0.13	0.2306	NA	NA	NA
	Male	80		NA	19:39		0.4616	NA	NA	NA
	Female	73		NA	13:30		0.4076	NA	NA	NA
	Male	82		NA	12:11		0.2530	NA	NA	NA
	Female	71		NA	12:00		0.3925	NA	NA	NA
PD	Female	87	80.8 ± 6.8	3	13:08	14:19 ± 0.12	0.2461	3/3	0	none
	Male	88		5	11:20		0.3607	5/3	0.5	moderate
	Female	75		3	14:40		0.6556	3/2	0	moderate
	Male	81		6	19:10		1.1524	6/2	3	moderate
	Female	73		3	13:20		0.4838	3/2	0	none

AD: Alzheimer’s disease; Ø Age ± SD: average age and standard deviation; DS: disease stage according to neuropathological classification (Braak scale); PMI: postmortem interval; Ø PMI ± SD: average PMI and standard deviation; NA: not applicable, CDR: clinical dementia rating, CERAD: Consortium to Establish a Registry for Alzheimer’s Disease.

**Table 2 cells-09-02580-t002:** Percentage intensity based absolute quantification (iBAQ) values of mitochondrial marker protein complexes. All samples show a similar percentage of both protein families, indicating comparable amounts of mitochondria in each sample. A detailed list of every protein assessed can be found in Appendix A.

Protein Family	CTRL 1134511	CTRL 12449/11	CTRL 3887/11	CTRL 6614/11	CTRL 6878/11	PD 10158/11	PD 10488/11	PD 11321/09	PD 11749/09	PD 4406/09
ATP Synthase	16.59	16.53	13.24	15.49	13.76	12.30	12.03	14.17	14.82	16.51
TIM/TOM complex	0.23	0.31	0.30	0.29	0.26	0.29	0.11	0.27	0.27	0.38

**Table 3 cells-09-02580-t003:** Differentially expressed proteins (overlap of MaxQuant and Spectral Counting analysis) and their annotated subcellular location.

Entry	Protein Names	Gene Names	Subcellular Location [CC]
**P21926**	**CD9 antigen (5H9 antigen) ***	***CD9 MIC3 TSPAN29 GIG2***	**Cell membrane**
O00142	Thymidine kinase 2, mitochondrial	*TK2*	Mitochondrion.
Q9BZE1	39S ribosomal protein L37, mitochondrial	*MRPL37 MRPL2 RPML2 HSPC235*	Mitochondrion
Q9BYT8	Neurolysin, mitochondrial	*NLN AGTBP KIAA1226*	Mitochondrion intermembrane space
P33316	Deoxyuridine 5′-triphosphate nucleotidohydrolase, mitochondrial	*DUT*	Nucleus
Q99798	Aconitate hydratase, mitochondrial	*ACO2*	Mitochondrion
P80404	4-aminobutyrate aminotransferase, mitochondrial	*ABAT GABAT*	Mitochondrion matrix.
Q9H6V9	Lipid droplet-associated hydrolase	*LDAH C2orf43*	Lipid droplet
P40926	Malate dehydrogenase, mitochondrial	*MDH2*	Mitochondrion matrix
P42126	Enoyl-CoA delta isomerase 1, mitochondrial	*ECI1 DCI*	Mitochondrion matrix
Q92665	28S ribosomal protein S31, mitochondrial	*MRPS31 IMOGN38*	Mitochondrion
O00330	Pyruvate dehydrogenase protein X component, mitochondrial	*PDHX PDX1*	Mitochondrion matrix.
P31930	Cytochrome b-c1 complex subunit 1, mitochondrial	*UQCRC1*	Mitochondrion inner membrane
Q96GW9	Methionine--tRNA ligase, mitochondrial	*MARS2*	Mitochondrion matrix
O14744	Protein arginine N-methyltransferase 5	*PRMT5 HRMT1L5 IBP72 JBP1 SKB1*	Cytoplasm

* bold: overrepresented in PD.

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
