# Peer review of "Proteomic Characterization of Synaptosomes from Human Substantia Nigra Indicates Altered Mitochondrial Translation in Parkinson’s Disease"

_cells, 2020, doi:10.3390/cells9122580_

Round 1

Reviewer 1 Report

The manuscript by Plum and colleagues describes the characterization of synaptosomes from human substantia nigra (SN) from 5 PD and 5 control subjects. The authors have identified 8 differentially expressed proteins (7 downregulated and 1 upregulated) in the SN of PD subjects. Most of these proteins appear to be related to mitochondrial function and highlight a deficit in this organelle in PD.

Major issues:

-       The manuscript requires professional language editing. It is very difficult to understand the results and the discussion. The manuscript will need to be reviewed following language editing.

-       Figure 6: The authors have used GAPDH as a loading control. The use of a protein as a loading control is to determine whether similar quantities of protein have been loaded in every well. As such, the signal should be as weak as possible so that one can detect variations in protein quantity. The strength of the GAPDH immunoreactive signal is so strong in this experiment that even if there was a variation in loading it would not be detectable. The ideal immunoreactive signal for a loading control would be that which was obtained for TK2. This western blot must be re-run in its entirety in order for the authors to make any comment regarding variation in protein expression. It is also unclear why the authors state “Due to the weak signal for TK2, no reliable quantification was possible”. The signal appears easily quantifiable.

-       Lns 432-435 – It is unclear why the authors have lined CD9 expression to BBB dysfunction.

- 

Minor Issues

-       Although Figure 4 is a novel presentation of the results, it is impossible to read some of the text. A color-coded key should be provided as a supplementary document.

-       Figure references on lines 234, 237 and 241 should be to Figure 2 (not figure 1).

-       When recording figures in the thousands, please use a comma as a separator rather than a period – e.g., Ln 265 should read 2,873 and 2,840 (rather than 2.873 and 2.840). Please correct all numbering.

Author Response

Reviewer 1:

Open Review

English language and style

(x) Extensive editing of English language and style required
( ) Moderate English changes required
( ) English language and style are fine/minor spell check required
( ) I don't feel qualified to judge about the English language and style

Yes

Can be improved

Must be improved

Not applicable

Does the introduction provide sufficient background and include all relevant references?

(x)

( )

( )

( )

Is the research design appropriate?

(x)

( )

( )

( )

Are the methods adequately described?

( )

(x)

( )

( )

Are the results clearly presented?

( )

( )

(x)

( )

Are the conclusions supported by the results?

( )

( )

(x)

( )

Comments and Suggestions for Authors

The manuscript by Plum and colleagues describes the characterization of synaptosomes from human substantia nigra (SN) from 5 PD and 5 control subjects. The authors have identified 8 differentially expressed proteins (7 downregulated and 1 upregulated) in the SN of PD subjects. Most of these proteins appear to be related to mitochondrial function and highlight a deficit in this organelle in PD.

Major issues:

  1. The manuscript requires professional language editing. It is very difficult to understand the results and the discussion. The manuscript will need to be reviewed following language editing.

The manuscript has been revised by a professional service regarding the language.

  1. Figure 6: The authors have used GAPDH as a loading control. The use of a protein as a loading control is to determine whether similar quantities of protein have been loaded in every well. As such, the signal should be as weak as possible so that one can detect variations in protein quantity. The strength of the GAPDH immunoreactive signal is so strong in this experiment that even if there was a variation in loading it would not be detectable. The ideal immunoreactive signal for a loading control would be that which was obtained for TK2. This western blot must be re-run in its entirety in order for the authors to make any comment regarding variation in protein expression. It is also unclear why the authors state “Due to the weak signal for TK2, no reliable quantification was possible”. The signal appears easily quantifiable.

We understand the reviewers’ point. It was our intention to further support the significance of the results from the comparative study. However, the amount of isolated synaptosomes was not sufficient to produce more than one Western blot. This allowed us to use only a single Western blot for the detection of all proteins one after the other, which definitely is not optimal. GAPDH was chosen as a loading control as it is usually used in Western blot experiments and as it was not found to be regulated in the differential study.

The reviews convinced us that the Western blot data is not optimal and we finally removed the data from the manuscript. The quantitative evaluation of the mass spectrometric data with two independent methods (spectral counting at MS2 level and MaxQuant intensity-based at MS1 level) itself demonstrates the significance of the results. We now additionally calculated the iBAQ value (Schwannhäusser et al, Nature 2011, https://doi.org/10.1038/nature10098) for all differential proteins which provides an accurate proxy for protein levels in every individual sample (see supplement Table 5). iBAQ values show the same trend for all differential proteins as the relatively quantitative values.

  1. Lns 432-435 – It is unclear why the authors have lined CD9 expression to BBB dysfunction.

Thanks for the comment. We added a respective sentence and reference here.

- Minor Issues

  1. Although Figure 4 is a novel presentation of the results, it is impossible to read some of the text. A color-coded key should be provided as a supplementary document.

We revised Figure 4 and included a more detailed figure in the supplement (supplement Figure 2).

  1. Figure references on lines 234, 237 and 241 should be to Figure 2 (not figure 1).

We thank the reviewer for pointing out this mistake. We changed the figure number.

  1. When recording figures in the thousands, please use a comma as a separator rather than a period – e.g., Ln 265 should read 2,873 and 2,840 (rather than 2.873 and 2.840). Please correct all numbering.

Thanks, we corrected all of them.

Reviewer 2 Report

In the manuscript entitled “Proteomic characterization of synaptosomes from human substantia nigra indicates altered mitochondrial translation in Parkinson’s disease” the authors present the results of a proteomics analysis of synaptosomes isolated from SN tissues of PD and control subjects. They highlight the downregulation of some mitochondrial proteins involved in protein synthesis in PD if compared to controls.

The content of the manuscript is of interest due to the lack of scientific literature about synaptosome proteomics in PD. However, results are not so well presented and convincing. In my opinion, major revisions are required to consider this manuscript for publication.

General comments:

- Even though I am aware of the difficulties on finding and working with patient-derived tissue samples, the number of subjects per group is very small (n=5). This is an actual issue if we consider that the PD group is not homogeneous, since PD, PDD and DLB have been considered as a single disease.

- As for the ProteomeXchange repository, the reviewer should be allowed to access and verify data (ask for reviewer credentials).

- The first paragraph of results is quite confusing and should be carefully revised for figure numbers; in addition, the main text should include a more detailed description of the overall strategy at the beginning. However, the main shortcoming of this paragraph is the lacking of a mitochondrial marker in the QC check of synaptosomes by Western Blot (figure 2D). Since the majority of proteins identified as differentially expressed by the proteomics analysis are mitochondrial proteins, it’s crucial to assess whether all synaptosomes preparations include “functional” synapses, with the presence of mitochondria. A Western blot analysis of the 15-23% Percoll fraction of all samples (n=10) showing the same expression of a mitochondrial marker protein (e.g., TIM, TOM, ATP-synthase) is crucial to avoid any bias in the quantification of mitochondrial proteins by proteomics. Otherwise the significance of the entire proteomics analysis will be mined.

- In the 3.2 section is not clear what the authors want to demonstrate by the PSM count. The fact that the PSM count is similar in all samples for the top 50 abundant proteins does not mean that these proteins are expressed in a comparable amount in all samples.

- In section 3.3 the authors show a nice graphical representation of the annotation clustering results. However, I strongly suggest the authors to perform an overrepresentation analysis on their protein list (n=584), so to identify statistically significant enriched pathways. This will actually demonstrate that the isolated synaptosomes are enriched in proteins that map to these structures. Finally, the STRING analysis as presented here is not so useful.

- In the 3.5 session a mitochondrial marker should be included (figure 6) to rule out the possibility that the reduction in the levels of the other proteins is merely due to the fact that synaptosomes from PD patients contain less functional mitochondria.

Specific comments:

Line 45: replace “SN” with “SNpc”.

Line 90: a brief explanation of the procedure should be included, not only referenced.

Table 1: the information about the disease type should be added (PDD, PD, DLB, sporadic, familiar, …).

Line 116: what does it mean that “protein concentration was estimated by amino acid analysis”? Which assay was used?

Line 120: the in-gel digestion protocol should be described in more details (lanes fractionation, protein:trypsin ratio, …).

Line 142: the repository is not yet publicly available (PXD005261), so I’d like to have reviewer credentials to access and verify the data.

Lines 163-168: it’s not so clear how the authors performed the “normalization” based on the master mixes.

Line 184: as in line 116, please specify the type of quantification assay employed.

Line 206: the procedure for the quantification of the fluorescent signals should be included at the end of the paragraph.

Line 209: a brief description of the strategy should be included in the main text (not only in the caption to figure 1).

Figure 1: I would also include WB in the figure (in addition to electron microscopy) as QC check after the density gradient centrifugation. Proteins separation through BisTris gel before tryptic in-gel digestion is lacking in both image and caption. In line 227 the authors say that only proteins identified by both algorithms and in ALL samples were accepted, while in M&M (line 160) they say that they accepted those present in 3 out of 5 subjects per group. Please clarify this point.

Line 211: replace “figure 2A” with “figure 2B”.

Line 213: replace “figure 2B” with “figure 2C”.

Lines 233-234: due to the low reproducibility of the procedure for the isolation of synaptosomes, all samples should be QC checked by WB and images should be included as supplementary material.

Line 234: replace “figure 1C” with “figure 2D”.

Line 237: remove “results are shown in figure 1C”.

Line 241: replace “figure 1C” with “figure 2D”.

Figure 3: I would move this figure to the supplementary material.

Figure 4: what is the meaning of the two stars in panel C?

Figure 5: unreadable. I would suggest to change the data presentation style.

Line 357: the supplementary material does not include this table.

Font size should be increased in some figures (difficult to read without zooming).

Some typo to be revised throughout the manuscript.

Author Response

Reviewer 2:

Open Review

English language and style

( ) Extensive editing of English language and style required
( ) Moderate English changes required
(x) English language and style are fine/minor spell check required
( ) I don't feel qualified to judge about the English language and style

Yes

Can be improved

Must be improved

Not applicable

Does the introduction provide sufficient background and include all relevant references?

(x)

( )

( )

( )

Is the research design appropriate?

( )

( )

(x)

( )

Are the methods adequately described?

( )

(x)

( )

( )

Are the results clearly presented?

( )

( )

(x)

( )

Are the conclusions supported by the results?

( )

( )

(x)

( )

Comments and Suggestions for Authors

In the manuscript entitled “Proteomic characterization of synaptosomes from human substantia nigra indicates altered mitochondrial translation in Parkinson’s disease” the authors present the results of a proteomics analysis of synaptosomes isolated from SN tissues of PD and control subjects. They highlight the downregulation of some mitochondrial proteins involved in protein synthesis in PD if compared to controls.

The content of the manuscript is of interest due to the lack of scientific literature about synaptosome proteomics in PD. However, results are not so well presented and convincing. In my opinion, major revisions are required to consider this manuscript for publication.

General comments:

  1. Even though I am aware of the difficulties on finding and working with patient-derived tissue samples, the number of subjects per group is very small (n=5). This is an actual issue if we consider that the PD group is not homogeneous, since PD, PDD and DLB have been considered as a single disease.

We agree that the study group was very small, which was due to the availability of the corresponding SN tissue in sufficient quantity. The classic motor symptoms of Parkinson's disease are caused by a degeneration of the SN and we therefore decided to include all subtypes.  Despite the limited number of specimen we found consistent protein changes with high significance and think our data is interesting for the scientific community. Especially as this is to our knowledge the first classical proteomic study of human synaptosomes in the context of PD.

  1. As for the ProteomeXchange repository, the reviewer should be allowed to access and verify data (ask for reviewer credentials).

We re-analysed and submitted all data to PRIDE. Data is accessible with the following details:

Project Name: Proteomic characterization of synaptosomes from human substantia nigra indicates altered mitochondrial translation in Parkinson’s disease
Project accession: PXD022092
Project DOI: 10.6019/PXD022092

Reviewer account details:
Username: [email protected]
Password: rFevO65f

  1. the first paragraph of results is quite confusing and should be carefully revised for figure numbers; in addition, the main text should include a more detailed description of the overall strategy at the beginning. However, the main shortcoming of this paragraph is the lacking of a mitochondrial marker in the QC check of synaptosomes by Western blot (figure 2D). Since the majority of proteins identified as differentially expressed by the proteomics analysis are mitochondrial proteins, it’s crucial to assess whether all synaptosomes preparations include “functional” synapses, with the presence of mitochondria. A Western blot analysis of the 15-23% Percoll fraction of all samples (n=10) showing the same expression of a mitochondrial marker protein (e.g., TIM, TOM, ATP-synthase) is crucial to avoid any bias in the quantification of mitochondrial proteins by proteomics. Otherwise the significance of the entire proteomics analysis will be mined.

Unfortunately due to low sample amounts were are not able to perform another Western blot. Moreover, the reviews convinced us to remove the Western blot data from the manuscript. Indeed, this comment is very valuable in order to improve the significance of our data. On the basis of our MS-data we calculated the iBAQ values (Intensity Based Absolute Quantification, Schwannhäusser et al, Nature 2011, https://doi.org/10.1038/nature10098) (providing an accurate proxy for protein levels) of ATP-Synthase and TIM/TOM associated proteins for each sample separately to ensure a similar presence of mitochondria in all samples. As stated in our revised manuscript iBAQ values are an option in MaxQuant analysis. iBAQ values are calculated using the raw intensities divided by the number of theoretical peptides, and are by that proportional to the molar quantities of each individual protein, resulting in an estimation of their absolute concentration. From the results it can be concluded that there is no bias for mitochondrial proteins. The results are summarized in Supplementary Table ST1.

  1. In the 3.2 section is not clear what the authors want to demonstrate by the PSM count. The fact that the PSM count is similar in all samples for the top 50 abundant proteins does not mean that these proteins are expressed in a comparable amount in all samples.

We are sorry, that our statement was not as clear as it should have been. We decided to revise this figure and incorporate a rank-intensity plot for the Top50 proteins of both CTRL and PD cases. For that LFQ intensities for each proteins were summed up for each group separately and proteins were ranked after highest intensity. This plot shall give information on similarities in expression levels of the most abundant proteins. We are well aware that the number of PSMs or the LFQ intensities are not directly comparable to the protein amount in all samples, but they do give hint on the protein distribution and expression pattern among the samples. With this plot, we are able to highlight that the Top 5 most abundant proteins are identical in both CTRL and PD cases, but that we do see differences in the Top 50 distribution in the higher rank proteins, proving differential expression patterns in CTRL compared to PD cases.

  1. In section 3.3 the authors show a nice graphical representation of the annotation clustering results. However, I strongly suggest the authors to perform an overrepresentation analysis on their protein list (n=584), so to identify statistically significant enriched pathways. This will actually demonstrate that the isolated synaptosomes are enriched in proteins that map to these structures. Finally, the STRING analysis as presented here is not so useful.

Thank you for the comment and suggestions. As recommended, we re-performed the annotation clustering with our core proteome data and decided to remove the STRING analysis.

  1. In the 3.5 session a mitochondrial marker should be included (figure 6) to rule out the possibility that the reduction in the levels of the other proteins is merely due to the fact that synaptosomes from PD patients contain less functional

We totally agree that PD patient possibly have less functional mitochondria, because of the lower number of neuronal cells compared to control subjects. Therefore, we compared similar protein amounts of PD and controls. A Western blot would not mirror differences based on different numbers of neuronal cells. Moreover, analysis of functional intact mitochondria in post-mortem human tissue is nearly impossible due to the preparation process as well as time between death and autopsy. We would also like to highlight that the mass spectrometry analysis did not reveal an overall change in mitochondrial proteins as demonstrated in the iBAQ calculations (mentioned above, point 3).

Specific comments:

Line 45: replace “SN” with “SNpc”.

We have changed the abbreviation.

Line 90: a brief explanation of the procedure should be included, not only referenced.

Thank you very much for this note. We added a paragraph in the manuscript (lines 93-102): Electron microscopic analysis was performed as described before [34].

Table 1: the information about the disease type should be added (PDD, PD, DLB, sporadic, familiar, …).

We added detailed information about the PD study group in table 1. Information about sporadic/familiar were not available.

Line 116: what does it mean that “protein concentration was estimated by amino acid analysis”? Which assay was used?

We changed the text accordingly. The determination of the peptide concentration was performed by amino acid analysis using the AccQ-Tag™ derivatization reagent. The method itself is described in detail in the reference [34].

Line 120: the in-gel digestion protocol should be described in more details (lanes fractionation, protein:trypsin ratio, …).

Due to the limited word count allowed by the journal we did not describe the method in detail within this manuscript. The detailed protocol is described in [34].

Line 142: the repository is not yet publicly available (PXD005261), so I’d like to have reviewer credentials to access and verify the data.

We re-analysed and submitted all data to PRIDE. Data is accessible with the following details:

Project Name: Proteomic characterization of synaptosomes from human substantia nigra indicates altered mitochondrial translation in Parkinson’s disease
Project accession: PXD022092
Project DOI: 10.6019/PXD022092

Reviewer account details:
Username: [email protected]
Password: rFevO65f

Lines 163-168: it’s not so clear how the authors performed the “normalization” based on the master mixes.

This part was removed from the manuscript.

Line 184: as in line 116, please specify the type of quantification assay employed.

See above.

Line 206: the procedure for the quantification of the fluorescent signals should be included at the end of the paragraph.

The Western Blot was removed.

Line 209: a brief description of the strategy should be included in the main text (not only in the caption to figure 1).

Thank you for the suggestion. We included a short description at the beginning of the results chapter.

Figure 1: I would also include WB in the figure (in addition to electron microscopy) as QC check after the density gradient centrifugation. Proteins separation through BisTris gel before tryptic in-gel digestion is lacking in both image and caption. In line 227 the authors say that only proteins identified by both algorithms and in ALL samples were accepted, while in M&M (line 160) they say that they accepted those present in 3 out of 5 subjects per group. Please clarify this point.

We changed this accordingly.

Line 211: replace “figure 2A” with “figure 2B”.

We changed the figure number.

Line 213: replace “figure 2B” with “figure 2C”.

The figure number was changed.

Lines 233-234: due to the low reproducibility of the procedure for the isolation of synaptosomes, all samples should be QC checked by WB and images should be included as supplementary material.

The reviewer is completely right: a quality check of each individual sample would be desirable. Indeed, the amount of protein extracted from the synaptosomal fractions of human subtantia nigra tissue is low especially for the disease cases. Therefore it was simply not practicable to carry out a quality check on each sample. Indeed, before starting this proteomic study of human synaptosomes we could verify the reproducibility of the purification as proven in Plum et al, J Proteomics 2013.

Line 234: replace “figure 1C” with “figure 2D”.

The figure number was changed.

Line 237: remove “results are shown in figure 1C”.

We have changed the wording.

Line 241: replace “figure 1C” with “figure 2D”.

The figure number was changed.

Figure 3: I would move this figure to the supplementary material.

We have once again completely revised Figure 3 to make it easier to understand and decided to leave it as part of the main manuscript. 

Figure 4: what is the meaning of the two stars in panel C?

We removed the stars in panel C.

Figure 5: unreadable. I would suggest to change the data presentation style.

Figure 5 was revised completely.

Line 357: the supplementary material does not include this table.

All supplementary files were uploaded through the Journals’ portal. Additionally, all supplementary data (except the figures) are included in the PRIDE upload.

Font size should be increased in some figures (difficult to read without zooming).

We changed the respective figures accordingly.

Some typo to be revised throughout the manuscript.

The manuscript was revised regarding language and typos.

Reviewer 3 Report

The manuscript deals with the proteomics characterization of SN synaptosomes. Authors suggest that results indicate altered mitochondrial translations.

I have many concerns about the conclusions drawn by the authors.

  1. By neglecting several prior art papers showing alterations in MRPs the authors tend to increase readers perceptions to novelty. There are few (4) papers that represent early proteomics characterization and none are cited. The same group published other papers on the same topic in the past 10 years.
  2. Experimental. I understand to obtain post-mortem samples, however an international collaboration with two brain banks should guarantee access to more specimens. Concerning protein identification, why using a 6-years-old database? Uniprot release 2014_10 was significantly different from the present one. How were ontologies associated to protein lists? Which over-representation analysis? Which correction for multiple testing? Also for univariate tests, comparing LFQ values, was a Benjamini-Hochberg correction applied?
  3. Results: Figure 2 is not completely original, and this should be acknowledged. Proteins altered between PD and controls have been filtered very strictly based on volcano plot criteria (significance and fold change). This is not a good choice for a subsequent over-representation analysis (ORA) because proteins that are changing not significantly (a common situation in a 5 vs. 5 comparison) could contribute to the identification of an enriched pathway. Actually, no ORA was performed, and no FDR is provided for enriched ontologies. Figure 6 does not show any evidence. This representative WB should be accompanied by a barplot with average plus/minus SD and a parametric test showing the calculated p-value. There is no evidence that DARS2 and neurolysin are changing significantly, nor there is any reason why TK2 cannot be quantified, whereas TK2 can. 
  4. Discussion: Single protein description makes no sense. IDs need to be analyzed by ORA so to give a FDR value to each pathway.

Author Response

Reviewer 3:

Open Review

English language and style

( ) Extensive editing of English language and style required
( ) Moderate English changes required
(x) English language and style are fine/minor spell check required
( ) I don't feel qualified to judge about the English language and style

Yes

Can be improved

Must be improved

Not applicable

Does the introduction provide sufficient background and include all relevant references?

( )

( )

(x)

( )

Is the research design appropriate?

( )

( )

(x)

( )

Are the methods adequately described?

(x)

( )

( )

( )

Are the results clearly presented?

( )

(x)

( )

( )

Are the conclusions supported by the results?

( )

( )

(x)

( )

Comments and Suggestions for Authors

The manuscript deals with the proteomics characterization of SN synaptosomes. Authors suggest that results indicate altered mitochondrial translations.

I have many concerns about the conclusions drawn by the authors.

  1. By neglecting several prior art papers showing alterations in MRPs the authors tend to increase readers perceptions to novelty. There are few (4) papers that represent early proteomics characterization and none are cited. The same group published other papers on the same topic in the past 10 years.

We have added some more references in the introduction part. However, even after an intensive literature search we could not find any other publications that investigated human synaptosomes in the context of PD or LBD in a classical proteomics approach. Therefore we really do consider our approach to be new. If, despite extensive searches, we have overlooked further relevant publications here, we would be very grateful for a specific reference so that we can acknowledge and reference these works accordingly.

  1. I understand to obtain post-mortem samples, however an international collaboration with two brain banks should guarantee access to more specimens. Concerning protein identification, why using a 6-years-old database? Uniprot release 2014_10 was significantly different from the present one. How were ontologies associated to protein lists? Which over-representation analysis? Which correction for multiple testing? Also for univariate tests, comparing LFQ values, was a Benjamini-Hochberg correction applied?

Thank you for pointing that out. We performed our proteomic analysis with the most recent SwissProt.fasta from uniprot (2020_10) and revised our findings. We did not apply correction for multiple testing as our sample size is comparable low and the biological variability of patient-derived samples has a tremendous effect on statistical significance. However, by choosing two different quantification-based strategies one based on MS1 quantification (MaxQuant) and one based on MS2 quantification (Spectral counting) we can ensure reliability of our significant proteins, by only choosing the overlap of both analysis strategies, resulting in 15 differential proteins.

  1. Results: Figure 2 is not completely original, and this should be acknowledged. Proteins altered between PD and controls have been filtered very strictly based on volcano plot criteria (significance and fold change). This is not a good choice for a subsequent over-representation analysis (ORA) because proteins that are changing not significantly (a common situation in a 5 vs. 5 comparison) could contribute to the identification of an enriched pathway. Actually, no ORA was performed, and no FDR is provided for enriched ontologies. Figure 6 does not show any evidence. This representative WB should be accompanied by a barplot with average plus/minus SD and a parametric test showing the calculated p-value. There is no evidence that DARS2 and neurolysin are changing significantly, nor there is any reason why TK2 cannot be quantified, whereas TK2 can.

Thank you for your suggestions concerning the ORA analysis. We revised the analysis and choose to incorporate proteins being significantly enriched in both analysis strategies (either MQ or SC). We analysed the two approaches separately using DAVID Bioinformatics Resources 6.8. This software allows to incorporate a background, which builds the basis of the enrichment analysis. As background we chose the human proteome dataset to ensure a correct identification of enriched pathways. We further on included the p-value and it’s correction, the FDR as well as the fold enrichment score all provided by DAVID in our Supplementary data, as well as partly in the presented Figure.

  1. Discussion: Single protein description makes no sense. IDs need to be analyzed by ORA so to give a FDR value to each pathway.

As mentioned in the comment above, FDR-values as well as fold enrichment scores and further statistics are now included in our Supplementary data and figures.

Round 2

Reviewer 1 Report

We thank the authors for considering comments made by this reviewer. The manuscript has been significantly improved and reads much better. The additional analysis and supplementary material have added weight to the manuscript.

There are a few minor edits that would improve the readability and presentation of the manuscript.

Ln 101 – Replace “described before” with ‘previously described’. Also replace “Shortly summarized” with ‘Briefly,’.

Section 2.2 should appear after section 2.4.

Ln 124 – Please change “All subjects died natural deaths” to ‘All subjects died of natural causes,’

Ln 145 – Please change “Previous” to ‘Prior’

Ln 147 – Please identify the LC system used, including make and model

Ln 153 - Orbitrap Elite mass spectrometer – please provide Manufacturer’s name

Ln 244 – it is unclear what “Summarized SN” refers to

Ln 257 – Please replace 10.500 with 10,500. Please make these changes to all numbering - they should appear in AP style.

Figure 2C – it would be beneficial if the image could be presented at a similar exposure as 2B

Ln 276 – Please replace “highly interesting differential candidates” with ‘unique candidates’.

Ln 321 – Avoid commencing a sentence with a number (e.g., 3.125). Please also use a comma to separate numbers in thousands (3,125) throughout the document.

Ln 324 – It is unclear what the following sentences means – “Of those 2.544 proteins, 362 proteins could be quantified in every sample.” What does ‘quantified’ mean?

Ln 330 – states “Proteins found in only one group were marked in red” but Ln 350 states “Proteins differing between CTRL and PD cases are marked in red”.

Which one is it? Please correct.

Figure 4. Some of the information cannot be read. A suggestion may be to list these again in a Supplementary Figure (linked to Figure 4) where the information is legible – similar to supplementary figure 1. For example for Figure 4C:

Human diseases (black)

Environmental information processing (cyan):

Genetic information processing (blue):

Metabolism (yellow): oxidative phosphorylation, amino acid metabolism, glycolysis……

Cellular processes (red): cytoskeletal proteins, exosomes, peroxisomes ……..

Organismal systems (pink):

Ln 400 – Please define PSM (as previous definition has been deleted)

Ln 411 – Please reference Table 3. Suggested text:

One protein was significantly over-represented (italic) and fourteen proteins were underrepresented (Table 3).

Ln 430 – delete the word ‘above’

Ln 445 – Please change “supporting our hypotheses raised” to ‘ supporting our original hypotheses’.

Ln 454 – Replace ‘the tricarboxylic acid cycle and the gluconeogenesis’ with ‘the tricarboxylic acid cycle and gluconeogenesis’.

Author Response

We thank all reviewers for their time and constructive input, which helped us to substantially improve our manuscript. In the following we will go into the respective comments point by point.

Reviewer 1

Open Review

English language and style

( ) Extensive editing of English language and style required
( ) Moderate English changes required
(x) English language and style are fine/minor spell check required
( ) I don't feel qualified to judge about the English language and style

Yes

Can be improved

Must be improved

Not applicable

Does the introduction provide sufficient background and include all relevant references?

(x)

( )

( )

( )

Is the research design appropriate?

(x)

( )

( )

( )

Are the methods adequately described?

(x)

( )

( )

( )

Are the results clearly presented?

( )

(x)

( )

( )

Are the conclusions supported by the results?

(x)

( )

( )

( )

Comments and Suggestions for Authors

We thank the authors for considering comments made by this reviewer. The manuscript has been significantly improved and reads much better. The additional analysis and supplementary material have added weight to the manuscript.

There are a few minor edits that would improve the readability and presentation of the manuscript.:

  1. Ln 101 – Replace “described before” with ‘previously described’. Also replace “Shortly summarized” with ‘Briefly,’
  2. Section 2.2 should appear after section 2.4.
  3. Ln 124 – Please change “All subjects died natural deaths” to ‘All subjects died of natural causes,’
  4. Ln 145 – Please change “Previous” to ‘Prior’
  5. Ln 147 – Please identify the LC system used, including make and model

We have made the corresponding changes to points 1-5 in the text.

  1. Ln 153 - Orbitrap Elite mass spectrometer – please provide Manufacturer’s name

We added the missing information in the mansucript.

  1. Ln 244 – it is unclear what “Summarized SN” refers to

The word „Summarized“ was removed.

  1. Ln 257 – Please replace 10.500 with 10,500. Please make these changes to all numbering - they should appear in AP style.

We changed this accordingly.

  1. Figure 2C – it would be beneficial if the image could be presented at a similar exposure as 2B

Unfortunately, this is not possible as we cannot modify the file without loss of quality.

  1. Ln 276 – Please replace “highly interesting differential candidates” with ‘unique candidates’.

We changed this accordingly.

  1. Ln 321 – Avoid commencing a sentence with a number (e.g., 3.125). Please also use a comma to separate numbers in thousands (3,125) throughout the document.

We changed this accordingly.

  1. Ln 324 – It is unclear what the following sentences means – “Of those 2.544 proteins, 362 proteins could be quantified in every sample.” What does ‘quantified’ mean?

A standard proteomic bottom up approach consists of several steps. First proteins are digested into peptides. Peptides are then identified by their amino acid sequence derived from the fragment ion spectra (MS2 level) of each peptide. The subsequent quantification can be carried out in two ways: Either on MS1 level, whereby the area under the MS1 peak is taken for the calculation of peptide quantity, or on MS2 level in which peptide spectrum matches (PSMs) are counted and summed. Lastly, peptides are infered to proteins using special Protein Inference Algorithms. Especially for low abundant peptides the quantification is not always possible, resulting in so called missing values. Additionally some peptides may not be present in both CTRL and PD cases. To ensure that our defined core proteome is not affected by disease, age, gender or algorithm we chose to only incorporate proteins, whose peptides could be quantified in every sample and both analysis strategies, resulting in 362 proteins.

We added two sentences in the text (version without track line 178 and lines 293/294) and changed “identified” to “quantified” in part 3.2. We hope that this point is clearer now.

  1. Ln 330 – states “Proteins found in only one group were marked in red” but Ln 350 states “Proteins differing between CTRL and PD cases are marked in red”.

Which one is it? Please correct.

Thank you for the comment, we changed the text to: “Proteins being in the Top 50 in group only, were marked in red (Figure 3)”, to clarify our statement

  1. Figure 4. Some of the information cannot be read. A suggestion may be to list these again in a Supplementary Figure (linked to Figure 4) where the information is legible – similar to supplementary figure 1. For example for Figure 4C:

Three additional figures were added in Supplemental Figure 1 in order to make ensure the legibility of the images.

  1. Ln 400 – Please define PSM (as previous definition has been deleted)

We have added the previous definition.

  1. Ln 411 – Please reference Table 3. Suggested text:

One protein was significantly over-represented (italic) and fourteen proteins were underrepresented (Table 3).

The table was referenced in line 369.

  1. Ln 430 – delete the word ‘above’

We changed this accordingly.

  1. Ln 445 – Please change “supporting our hypotheses raised” to ‘ supporting our original hypotheses’.

We changed this accordingly.

  1. Ln 454 – Replace ‘the tricarboxylic acid cycle and the gluconeogenesis’ with ‘the tricarboxylic acid cycle and gluconeogenesis’.

We changed this accordingly.

Reviewer 2 Report

The authors did a great work to improve clarity and significance of the results presented in the manuscript.

Data repository is now accessible for revision. I appreciated that the authors re-analysed their data using a 2020 reference database.

I am convinced that the quantification of mitochondrial marker proteins adds some value to the work, ruling out the possibility that the quantitative change of mitochondrial proteins in synaptosomes was due to technical biases in sample preparation. I wouldn’t have removed the Western blot analysis from the results, but I understand the fact that protein extracts were no more available to be loaded again and checked for marker mitochondrial proteins.

Figures are now clearly readable.

I suggest the authors to carefully check some contents that can be moved, in my opinion, from results to M&M. However, I appreciate the presence of a much more detailed description of the methods in general.

Also, I would avoid a so detailed description of the function of the significantly changing proteins in the discussion section. I would only highlight why these could be interesting in the context of PD.

In conclusion, I think that the work can be accepted for publication after minor revisions.

Author Response

We thank all reviewers for their time and constructive input, which helped us to substantially improve our manuscript. In the following we will go into the respective comments point by point.

Open Review

English language and style

( ) Extensive editing of English language and style required
( ) Moderate English changes required
(x) English language and style are fine/minor spell check required
( ) I don't feel qualified to judge about the English language and style

Yes

Can be improved

Must be improved

Not applicable

Does the introduction provide sufficient background and include all relevant references?

(x)

( )

( )

( )

Is the research design appropriate?

(x)

( )

( )

( )

Are the methods adequately described?

( )

(x)

( )

( )

Are the results clearly presented?

( )

(x)

( )

( )

Are the conclusions supported by the results?

( )

(x)

( )

( )

Comments and Suggestions for Authors

The authors did a great work to improve clarity and significance of the results presented in the manuscript.

Data repository is now accessible for revision. I appreciated that the authors re-analysed their data using a 2020 reference database.

I am convinced that the quantification of mitochondrial marker proteins adds some value to the work, ruling out the possibility that the quantitative change of mitochondrial proteins in synaptosomes was due to technical biases in sample preparation. I wouldn’t have removed the Western blot analysis from the results, but I understand the fact that protein extracts were no more available to be loaded again and checked for marker mitochondrial proteins.

Thank you very much for this view. We also agree that the Western blot data could have been supportive.  Indeed, as the exact quantification was a challenge due to the limited amount of tissue/protein we thought it is more scientifically correct to delete these data from the manuscript.  

Figures are now clearly readable.

  1. I suggest the authors to carefully check some contents that can be moved, in my opinion, from results to M&M. However, I appreciate the presence of a much more detailed description of the methods in general.

We made some changes in the M&M part.

  1. Also, I would avoid a so detailed description of the function of the significantly changing proteins in the discussion section. I would only highlight why these could be interesting in the context of PD.

We appreciate this recommendation and think this is a matter of taste. To our experience it is beneficial to have a brief description of the protein candidates within the manuscript underlined with references to the appropriate manuscripts. Thereby, the readers can evaluate by themselves if the protein is of interest for their research.

In conclusion, I think that the work can be accepted for publication after minor revisions.